# Mitochondrial Dysfunction in Parkinson’s Disease: A Contribution to Cognitive Impairment?

**DOI:** 10.3390/ijms252111490

**Published:** 2024-10-25

**Authors:** Antonella Scorziello, Rossana Sirabella, Maria Josè Sisalli, Michele Tufano, Lucia Giaccio, Elena D’Apolito, Lorenzo Castellano, Lucio Annunziato

**Affiliations:** 1Department of Neuroscience, Division of Pharmacology, Reproductive and Dentistry Sciences, School of Medicine, Federico II University of Naples, Via Pansini 5, 80131 Naples, Italy; rossana.sirabella@unina.it (R.S.); michele.tufano@unina.it (M.T.); lgiacciol@gmail.com (L.G.); elena.dapolito@unina.it (E.D.); lorenzo.castellano@unina.it (L.C.); 2Department of Translational Medicine, Federico II University of Naples, 80138 Napoli, Italy; mariajose.sisalli@unina.it; 3IRCCS Synlab SDN S.p.A., Via Gianturco 113, 80143 Naples, Italy

**Keywords:** Parkinson’s disease, mitochondria, cognitive impairment, synaptic dysfunction, α-synuclein neurons

## Abstract

Among the non-motor symptoms associated with Parkinson’s disease (PD), cognitive impairment is one of the most common and disabling. It can occur either early or late during the disease, and it is heterogeneous in terms of its clinical manifestations, such as Subjective Cognitive Dysfunction (SCD), Mild Cognitive Impairment (MCI), and Parkinson’s Disease Dementia (PDD). The aim of the present review is to delve deeper into the molecular mechanisms underlying cognitive decline in PD. This is extremely important to delineate the guidelines for the differential diagnosis and prognosis of the dysfunction, to identify the molecular and neuronal mechanisms involved, and to plan therapeutic strategies that can halt cognitive impairment progression. Specifically, the present review will discuss the pathogenetic mechanisms involved in the progression of cognitive impairment in PD, with attention to mitochondria and their contribution to synaptic dysfunction and neuronal deterioration in the brain regions responsible for non-motor manifestations of the disease.

## 1. Introduction

Parkinson’s disease (PD) is the second most common neurodegenerative condition in the world after Alzheimer’s disease (AD), and its rapid global growth has taken on worrying dimensions in recent years compared to the pandemic, except for its non-infectious characteristics [1]. From a neuropathological point of view, PD is characterized by the demise of dopaminergic (DA) neurons in the *substantia nigra pars compacta* (SNc) and the reduction in (DA) tone at the level of the *striatum* [2]. Specific neuropathological hallmarks of disease are represented by the presence of intraneuronal protein aggregates called Lewy bodies and Lewy neurites, which are eosinophilic cellular inclusions comprising a dense core of filamentous material that mainly consists of α-synuclein [3]. The presence, number, and specific anatomical localization of Lewy bodies as well as Lewy neurites are used for the diagnosis of PD [4,5]. However, PD is characterized by more widespread pathology in other brain regions and involves non-dopaminergic neurons as well. It is not yet clear what causes are related to the disease onset. However, genetic and environmental factors as well as their interactions have been claimed to be pathogenic in disease development and progression [6]. Two forms of PD have been described: the sporadic form, which occurs in approximately 95% of cases, and the familial form, which occurs in the remaining 5% of cases and recognizes a genetic link [7]. Although PD is age related and its incidence increases with age, it is wrong to think that it affects only the elderly, as there is a large percentage of people whose age of onset is lower than 60 years or even 50 years [8].

Over the centuries, numerous scientists have contributed to the clinical and pathological description of this disease, although a complete understanding remains elusive. PD poses a significant challenge because of its multifactorial nature and diverse spectrum of molecular and cellular alterations coupled with clinical manifestations that affect disease progression [6]. Moreover, the inaccessibility of the brain exacerbates the difficulty in studying brain function during disease development and formulating a diagnosis in the early stages [7]. As a result, the more we study PD, the more we recognize its intricate and heterogeneous nature, which makes rigorous classification increasingly difficult, as seen in many other degenerative diseases. In this regard, mitochondrial dysfunction, a mechanistic condition related to prodromal symptoms that lead to early deficits in cognition in PD patients, would represent a useful and novel approach to figure out how drugs that are able to counteract mitochondrial dysfunction might be useful in slowing down neuronal degeneration in the brain regions that are tightly related to neurotransmitter dysfunction. This hypothesis is further supported by the emerging evidence associating mitochondria with synucleinopathies [9]. Indeed, mitochondria, which are considered the powerhouse of the neurons, provide the energy required for neuronal excitability, synaptic activity, and plasticity through mitochondrial oxidative phosphorylation (OXPHOS) [10]. Moreover, they are dynamic organelles that traverse long distances to meet spatiotemporal ATP requirements in more demanding regions of the neuron. However, these mechanisms can become defective in PD in many of the brain regions involved in the control of cognition, attention, mood, memory, and executive functions [11,12,13,14,15,16,17,18,19], as reported in Table 1.

This review points to a thorough discussion of the intricate relationship between cognitive decline and PD, focusing on the underlying factors that contribute to the disorder and the brain regions involved, as well as including the role of mitochondria in synaptic plasticity, which is instrumental in cognition and behavior. We believe that recognizing the distinctive traits of cognitive impairment, which is one of the most frequently observed non-motor symptoms associated with the emergence of motor dysfunction in patients with PD, is of significant importance, as it might become a key component in formulating an early diagnosis. This is because identifying cognitive decline before extensive damage to dopaminergic neurons occurs enables the implementation of therapeutic interventions at an earlier stage, potentially slowing the progression of the disease.

## 2. Cognitive Impairment in PD

### 2.1. Clinical Symptoms of Cognitive Dysfunctions

The clinical manifestations of cognitive impairment associated with PD include a wide spectrum of symptoms ranging from Social Cognitive (SC) impairment, which occurs frequently in the early stage of the disease also in the absence of any other cognitive disorder [20], and Mild Cognitive Impairment (MCI), which appears in the early stages of the disease and is present in 10–20% of patients at the moment of PD diagnosis, to severe dementia, such as Parkinson’s Disease Dementia (PDD), which affects up to 90% of patients in the late stages [21]. Regarding the spectrum of symptoms accompanying cognitive dysfunction in PD, SC impairment has been recently identified as an early prodromal symptom in newly diagnosed PD patients, with features that make it different from MCI [20]. SC impairment represents the attitude of how to process, store, and apply information regarding interpersonal relationships and complex social interactions [22,23]. It depends on the consciousness of the other’s thoughts and intentions; social problem-solving abilities; understanding metaphors, humor, and sarcasm; or making moral decisions [22].

To date, the link between the time of occurrence and the extent of impairment in affective and cognitive manifestations of the SC domain in PD patients are under investigation [24,25]. Similarly, the relationship with other cognitive functions like mood [26,27,28], behavior [29], pharmacological treatments [27,30,31], and motor severity [25,26,27,31,32] remains to be clarified.

Boller and co-workers also found deficits in both visuoperceptive and visuomotor tasks that were independent of intellectual deficit [33]. In addition, patients with the greatest impairment of motor function display the greatest visuospatial deficit, suggesting that shortages in perceptual motor tasks may be, in part, responsible for problems in the sequential organization of behavior. Conversely, instrumental functions, such as language and ordinary activities, are less impaired in PD patients, until dementia reaches its stage of global cognitive decline, compared with the same symptoms occurring in AD patients, in which they appear more pervasive during all stages of neurodegeneration [8,34,35].

Other symptoms that coexist in PD patients with cognitive impairment include depression, hallucinations, and sleep disturbances [36,37,38]. Interestingly, depression associated with PD is milder than the endogenous form, appears in 35% of patients affected by PD, and is associated with changes in the activities of serotonin and norepinephrine neurons as well as with the impairment of cholinergic fibers, apart from dopamine pathways. Moreover, as reported for other non-motor symptoms, depression may precede the onset of motor dysfunction, and it is correlated with disease duration, severity of motor symptoms, and fluctuations in drug response. Unfortunately, depression associated with PD is a complex pathological phenomenon in terms of diagnosis, since it may be a consequence of PD, a reaction to the disability occurring in PD, a separate disease, or a combination of the above-mentioned conditions. This aspect is further worsened by the fact that many of the non-motor symptoms, including cognitive decline, sleep disorders, psychosis, and autonomic symptoms, may contribute to the increased probability of developing depression in PD [39,40,41,42]. Hallucination and sleep disorders are also included among the non-motor symptoms that characterize psychosis and may accompany PD in the later stage of illness in 40% of patients. In addition, the pharmacological treatments based on L-DOPA or dopamine agonists in PD patients might be responsible for psychosis, thus explaining the pathogenesis of these symptoms [43].

Conventionally, two types of dementia have been identified in patients with PD. The first type, known as “Lewy body dementia” (LBD), is characterized by the appearance of both cognitive and motor decline occurring at the same time or within a year of disease diagnosis [44,45]. The other type, known as PDD, is characterized by the onset of dementia that arises later in PD diagnosis. Despite the differences in the temporal sequence, the two forms of dementia are similar in terms of neuropathological and clinical features [46]. In both forms, the damage of dopaminergic neurons in the *substantia nigra*, which is characteristic of PD, is associated with considerable cholinergic and noradrenergic neuronal depletion in the pedunculopontine nucleus and in the *locus coeruleus* (LC), respectively [47,48]. In this regard, it is worth mentioning that dopamine depletion in the canonical nigrostriatal pathway is mainly responsible for impairments in working memory, planning and sequencing, task switching, response inhibition, memory recall, verbal fluency, and psychomotor speed, all of which are aspects related to the MCI condition [22,49].

On the other hand, impairments in acetylcholine, noradrenaline, and serotonin neurons are implicated in amnestic memory, language, and visuospatial impairments, mainly indicating a transition to the PDD phenotype [49,50]. These findings are in line with the dual-syndrome hypothesis proposed to explain the differences observed in the cognitive phenotype and progression of PD. Accordingly, two main neuropathological and neurochemical correlates have been identified. The first is associated with a greater frontal–striatal network dysfunction consequent to dopamine depletion and is responsible for deficits in attention, working memory, planning, and response inhibition; the second is associated with wide posterior cortical degeneration, which mainly involves cholinergic depletion and is responsible for dementia. Interestingly, these two forms are also associated with a difference in temporal appearance and evolution, since the first can be present early in the course of disease or can be associated with a low risk of progression over time; the second form appears later and is instead committed to evolve toward dementia [51]. From a neuropathological point of view, these two forms are associated with abnormal protein aggregates composed of Lewy bodies and neurites, beta-amyloid plaques, and neurofibrillary tangles, as confirmed by post-mortem studies in different brain regions of patients with PD [52,53,54]. More interestingly, neuroimaging studies in PD patients with MCI or dementia have revealed a good relationship between protein aggregates, anatomical alterations in grey matter (GM) and white matter (WM), and impairment in distinct cognitive domains, confirming that both frontal and posterior functional connectivity changes may be considered as possible biomarkers of cognitive impairment progression. Specifically, the analysis of structural brain alterations performed in patients with different levels of cognitive impairment, although heterogeneous, reveals progressive temporo-parietal thinning and frontal atrophy that might be considered as hallmarks of progression towards dementia [52]. Conversely, by examining WM alterations across the full spectrum of cognitive status detected in PD patients, a more consistent and unique pattern of modifications underlying an impairment in distinct cognitive domains can be observed [55,56]. In line with these results, it has been hypothesized that PD alterations primarily result from synaptic dysfunction, mainly due to the presence of Lewy bodies and Lewy neurites, which subsequently evolve into neuronal cell death [54,57]. This hypothesis underlines the relevance of neuroimaging in detecting cognitive correlates as biomarkers for PD-related cognitive impairment diagnosis. However, considering the neuropathological, neurochemical, and neuropsychological heterogeneities of cognitive profiles, studies combining neuroimaging and sensitive cognitive tests are mandatory to clarify the interplay between the neurodegenerative processes and the structural disconnections in brain functional networks that are responsible for the development of dementia. Indeed, although cognitive impairment in PD represents a continuous process that affects almost all patients, attempts to create a temporal classification of cognitive decline in SC domain, MCI, and PDD have been unsuccessful due to the heterogeneity of symptoms described in each group of patients [58]. Moreover, patients with PD display transient cognitive disturbances that are probably correlated with other pre-existing pathological conditions or pharmacological treatments, which may contribute to the complexity of the diagnosis and should not to be confused with dementia [35]. In this regard, it is also worth mentioning the impact of infections, since a variety of pathogens from viruses and bacteria to fungi and protozoa have been detected in the CNS using histochemical and molecular biology methods [59]. On the other hand, changes in the gut–brain axis can also affect the CNS via the secretion of neurotransmitters, vitamins, important fatty acids such as butyrate, and misfolded proteins like α-syn by gut microbes [60]. Interestingly, not all infections lead to neurodegeneration; therefore, it is possible to speculate that genetic susceptibility factors may help to explain why some people are more vulnerable to the detrimental effects of infections than others. However, infections are only one possible cause for transient cognitive disturbance in PD patients. Indeed, other factors, such as metabolic disease, and lifestyle aspects, including diet, sleep, exercise, stress, and smoking, might also play a role. During recent years, it also emerged that dehydration may contribute to mortality in PD [61]. Moreover, the results of a meta-analysis published in 2020 demonstrated a significant association of both hypothyroidism and hyperthyroidism with an increased risk of PD development. This is largely related to the effect of thyroid hormones on the function of dopaminergic neurons, and it is dependent on the induction of nuclear receptor-related 1 (nurr1) protein expression [62], which is essential for the survival and function of mesencephalic dopaminergic neurons. Indeed, the thyroid hormone modulates the expression of enzymes and transporters that are important for dopamine synthesis and storage, including tyrosine hydroxylase, dopamine transporter, vesicular monoamine transporter 2, and L-aromatic amino acid decarboxylase [63]. Therefore, it is expected that thyroid hormone deficiency may lead to a decrease in the number and function of mesencephalic dopaminergic neurons. On the other hand, thyroid hormone excess can lead to increased cellular metabolism, resulting in a high burden of oxidative stress from the mitochondria in multiple tissues, including neurons [64,65]. In addition, a relationship between thyroid disease and the risk of developing familial forms of PD have been described, suggesting that the two diseases may share a common genetic predisposition [66].

Another interesting aspect that needs to be mentioned with the aim of identifying the causes of transient cognitive decline in PD is related to nutrition and vitamin supports. The deficiency of several macronutrients and micronutrients represents a common condition in PD patients due to concomitants pathologies like depression, cognitive impairment, gastrointestinal dysmotility, and dysphagia [67], which, in turn, cause reduced food intake. Among them, pyridoxine (vitamin B 6) is an essential micronutrient available widely through dietary intake and synthesis by gut microbiota. Functional deficiency of pyridoxine in PD can be attributed to decreased intake, impaired absorption, and interaction with dopaminergic medications [68]. Finally, we cannot overlook the growing bulk of evidence related to the contribution of genetics to cognitive dysfunction in PD. It has been reported that monogenic forms of genetic PD show distinct cognitive profiles and an increased rate of cognitive decline progression [69]. Indeed, cognitive impairment is higher in GBA- and SNCA-associated PD, lower in Parkin- and PINK1-PD, and possibly milder in LRRK2-PD. Specifically, recently described data demonstrate the increasing risk of early dementia in patients with a mutation in the SNCA gene, like SNCA triplications and the p.E46 K mutation, whereas among LRRK2-PD patients, the frequency of cognitive impairment is similar or lower than that observed in idiopathic PD. Similarly, recessive forms of familial PD, including Parkin-PD and PINK1-PD, are generally characterized by a lower frequency of cognitive impairment, while GBA gene mutations, another risk factors for PD, have shown an increased probability of developing dementia [69]. These variabilities in the genetic forms of PD further contribute to explain the different cognitive involvements in each form of genetic PD.

All these potential pathogenetic conditions related to cognitive decline in PD further contribute to the heterogeneity of the populations studied and make it difficult to formulate a classification of progressive forms of dementia. Moreover, these limitations negatively impact the design and interpretation of pharmacological trials both in terms of symptomatic and pharmacological effects. Therefore, to overcame this gap is extremely important not only for the diagnosis and prognosis of the disease but also for the treatment of symptoms related to PD-associated cognitive decline, since, till now, many of the treatments employed were inefficacious [70].

### 2.2. Limits in Clinical Studies Dealing with Cognitive Impairment in PD

Until a few years ago, studies on PD have mainly focused on the mechanisms underlying motor disorders, whereas less attention has been paid to non-motor features, which are widely recognized as important factors in disease-related disabilities. This was due to a lack of attention to the anatomical and functional connections between the basal ganglia and prefrontal cortex, which is a neuronal pathway that is responsible for its involvement in cognition and memory. Therefore, the decrease in information flow through these circuits leads to cognitive dysfunctions, which, among the non-motor symptoms, are the most common in PD (Figure 1). Moreover, considering that dopaminergic neurons influence both the meso–cortico-limbic and sensory–motor systems, patients with PD may experience cognitive dysfunctions in addition to frontal lobe executive dysfunctions [71].

Although cognitive impairment affects a significant percentage of individuals, even in the early stages of the disease, its mechanistic aspects are still under investigation. This is mainly due to the heterogeneity of symptoms, ranging from executive dysfunction to difficulties in attention, mental processing in learning new information, and memory impairments that include working memory, long-term memory, visuospatial memory, and procedural learning [72], which may also occur during all stages of disease.

Therefore, clinical trials aimed at understanding the molecular mechanisms leading to cognitive impairment in PD at different stages of disease present some limitations mainly due to variability in patient responses and the lack of standardized methodological approaches [73]. Another important obstacle emerging from the analysis of clinical trials performed till now is also related to the differences in targeting a single or multiple domains of cognitive functions that accompany PD [74]. Therefore, a systematic revision of trials describing cognitive-based interventions in people with PDD and PD-MCI has been recently proposed with the aim of unifying the terminology and the methods of investigation applied in the studies to set up some sort of useful guidelines to properly approach the study of cognitive dysfunctions in PD both in terms of diagnosis and treatment [74].

This is extremely important, as the cognitive deficits observed in PD patients are different to those observed in other dementias, and therefore, interventions may need to be tailored to the cognitive domains commonly affected in PD. The main feature of dementia observed in PD is the impairment of executive functions, consisting in the ability to process and recall information from memory, as well as to plan, organize, and regulate goal-directed behaviors [75,76,77]. Therefore, these aspects need to be considered when a clinical trial has to be planned in order to use the appropriate tests to assess the type and the severity of deficits, thus minimizing the variations in study designs, outcome measures, and intervention protocols, which still present a difficulty in directly comparing the efficacy of these trials.

### 2.3. Cognitive Impairment in Animal Models of PD

In the light of the above considerations, animal models mimicking the pathophysiological features occurring in PD represent a valuable tool for studying the mechanisms of disease development in humans [78]. These include genetic (transgenic) and non-genetic (neurotoxic) models, which reproduce familial and sporadic diseases, respectively [79,80]. Among these, the most common were (1) rodents and primates intracerebrally administered MPTP, 6OH-DA, or preformed α-syn fibrils [47,81,82,83] as a model of dopaminergic impairment in the nigrostriatal pathway and (2) transgenic mice bearing human mutations [80,84,85,86].

The two toxins MPTP and 6OH-DA are introduced directly into the brain, with different effects on dopaminergic neuronal depletion depending on the region of administration. Indeed, a significant and sudden loss of TH-positive neurons can be detected in mice in which the toxins are administered into the SNc, whereas a slow neuronal impairment, also occurring after a few weeks, can be observed if the toxins are intracerebrally injected into the *striatum* [85]. This is extremely advantageous because it mirrors the slower development of motor impairment that is typical of PD and provides the opportunity to effectively analyze cognitive deficits as well as other non-motor disorders that occur earlier in PD. In this regard, Tadaiesky et al. demonstrated that PD reproduced in rats via bilateral infusions of 6-OHDA into the *striatum* is associated with reduced sucrose consumption and increased performance time in the water maze test [87]. These alterations reflect the clinical profiles of depression, anxiety, and impaired cognitive function. Similarly, the introduction of 6-OHDA or MPTP into the SNcs of rats causes a spatial memory deficit in the water maze test [88,89]. In addition, new animal models of PD have been developed by inoculating preformed α-syn fibrils (PFF) in the *substantia nigra* or the *striatums* of rodents [90]. These models, mimicking the ectopic expression or overexpression of the protein α-syn, allow for the replication of the pathological hallmarks of PD (Figure 2). Indeed, the use of these models importantly contributed to study the roles of toxic forms of α-syn in neurodegeneration and their influence in the processes occurring before cell death, such as changes in synaptic transmission, basal ganglia plasticity, and mechanisms underlying motor learning [91,92].

Although animal models of toxin-induced PD have been widely validated, they still produce highly variable results depending on the area into which the toxin is injected. Therefore, the discovery of familial mutations in 10% of patients with PD prompted neuroscientists to generate several genetic models for studying the disease with the aim of addressing the current disadvantages of using neurotoxins. The main advantage of transgenic models over neurotoxins is that they cause a mild and progressive onset of the disease, which is more congruent with PD development, providing advantages for mapping the etiology of possible deficits through cognitive function tests in the early stages of PD [93]. In this scenario, mice bearing α-syn mutations are commonly used to investigate the role of fibril-forming mutant α-syn in the pathogenesis of cerebral damage, such as in the spinal cord throughout the brainstem, the deep cerebellar nuclei, deep cerebellar white matter, and some regions of the thalamus, such as the medioventral, ventromedial, and paracentral nuclei [84]. Interestingly, all of these animal models have shown that α-synucleinopathy and mitochondrial dysfunction are detectable in the different brain regions involved in cognition depending on the age, further confirming that the aggregation of toxic α-syn forms may cause mitochondrial dysfunction and *vice versa*. Moreover, other genes have been identified as causative or contributory factors in PD development, including leucine-rich repeat kinase 2 (LRRK2) and parkin (PRKN) [86]. Common variants in these genes are also associated with the development of cognitive decline [94]

To date, a multitude of tests has been used to study cognitive function, memory and learning in PD models, as well as the impairment of mitochondrial function. They can be applied in vivo as the Morris water maze test, passive and active avoidance tests, episodic memory and object recognition tests, and social recognition tests [95,96] or ex vivo for mitochondrial oxygen consumption, ATP production, free radical generation, and loss of antioxidant defenses [97].

In addition to its value in determining the pathogenetic mechanisms leading to neuronal degeneration in PD as well as the assessment of cognitive impairment as a prodromal symptom of motor dysfunction, which is the only clinical feature for PD diagnosis at the moment, the availability of animal models that better resemble the early stage of clinical PD is very promising for investigating new therapeutic strategies focused on counteracting mitochondrial dysfunction and synucleinopathy in dementia associated with PD. In this regard, the development of preclinical models that can accurately reflect the neurobiology of PD and PDD in humans is of paramount importance for preventing dopaminergic neuronal death and halting PD progression [98,99,100,101,102].

## 3. α-Synuclein and Mitochondrial Involvement in the Pathogenetic Mechanisms Related to Cognitive Decline in PD

The cellular and molecular mechanisms implicated in the cognitive impairment associated with PD embrace a series of pathological features, including changes in the conformational structure of synaptic proteins, mitochondrial dysfunction and oxidative stress, and metabolic disturbances [11,22,23,24]. These effects are involved in the complex dynamics of neuroplasticity rearrangements in the framework of ongoing neurodegenerative processes (Figure 2), which strongly reflects the state of cognitive reserve of PD patients in the early and advanced stages [103,104]. In this scenario, the pathological deposition of α-syn and mitochondrial dysfunction appear to play a key role in cognitive impairment occurring in PD due to the causative relationships among bioenergetic demand, axonal arbor size, and the vulnerability of highly branched dopaminergic *nigro-striatal* axons [105] Indeed, experimental conditions that can impair mitochondrial complex I activity and increase α-syn deposition within mitochondria by modifying the synaptic activity may contribute to cognitive dysfunction [106,107,108]. Moreover, α-syn accumulation in the hippocampus and cortex receiving dopaminergic projections from the midbrain further contributes to the development of early manifesting cognitive and neuropsychiatric deficits [75]. Finally, the idea that abnormalities in mitochondrial function are a critical step in initiating dopaminergic neuronal dysfunction, leading to parkinsonian phenotypes and, ultimately, to dementia associated with Lewy bodies (LBD), is also supported by the pathophysiological features displayed by genetic forms of PD, such as those associated with LRRK2, PRKN [86], and α-synuclein (SNCA) gene mutations, in which cognitive impairment is also described [109].

Specifically, in this last case, dysfunctional mitochondrial bioenergetics were followed by a decrease in the number of DA terminals and morphological and ultrastructural alterations that occurred early in the course of the disease [110,111]. In the following Section 3.1 and Section 3.2, the pathogenetic involvement of abnormal α-syn accumulation and mitochondrial dysfunction in cognitive decline associated with PD will be discussed in more detail.

### 3.1. Role of α-Syn

α-syn is an acidic protein of 140 amino acids whose normal function within neurons is not fully understood. However, it appears to be involved in synaptic vesicle trafficking and mitochondrial function [47,99,112,113,114,115,116]. It is concentrated at the presynaptic level, where it is thought to mediate neurotransmitter release, especially in cells with pacemaker activity, such as dopaminergic neurons of the substantia *nigra pars compacta* (SNpc), the neuronal population that is most severely affected in PD [117]. On the other hand, in vivo and in vitro studies also demonstrated that α-syn may likewise be involved in the alteration of synaptic homeostasis by inducing detrimental effects at the postsynaptic level [118]. This last finding is dependent on the interference of α-syn with glutamatergic neurotransmission not only at the level of dopaminergic striatal terminals, where it reduces dopamine release, but also at level of postsynaptic density of medium spiny neurons by interfering with different subunits of NMDA receptors, respectively [119,120,121,122,123,124]. Moreover, α-syn may also interact with the NMDA receptors expressed by cholinergic interneurons in the *striatum*. Through these mechanisms, α-syn modifies the synaptic structure and blocks long-term potentiation in both dopaminergic and cholinergic neurons, causing either motor or behavioral alterations [91,122]. Hallam and collaborators demonstrated that the impairment in NMDAR activity, which is caused by aberrant insoluble and phosphorylated α-syn deposits at synaptic terminals, temporally correlates with increased nitric oxide synthesis and S-nitrosylation of the dendritic scaffold protein, microtubule-associated protein 1A [125]. These data suggest that the loss of synaptic function in Lewy body dementia may result from synucleinopathy-evoked nitrosative stress and subsequent NMDAR dysfunction [125].

Structurally, α-syn has three modular regions: (a) the N-terminal region, characterized by amphipathic repeats that tend to form an α-helix structure, which is responsible for membrane interactions; (b) the central hydrophobic region (NAC), which represents the non-amyloid β-component that displays aggregation domains; and (c) the acidic C-terminal region involved in Ca^2+^ binding and chaperone-like activity [126]. Through its N-terminal region, α-syn is stably anchored to the lipid surface in the presence of low Ca^2+^ concentrations, whereas at higher Ca^2+^ concentrations, the NAC and C-terminal regions also exhibit lipid-binding properties [126]. Under physiological conditions, α-syn exists in different conformations in equilibrium between unstructured soluble monomeric and tetrameric forms. Under pathological conditions, α-syn aggregates into oligomers, protofibrils, and fibrils, leading to the formation of protein inclusions known as Lewy bodies [3]. In this regard, it is important to underline the role of cellular environments in influencing the hydrophobic and electrostatic intermolecular interactions that control the solubility of the monomeric form of α-syn, with consequent alterations responsible for the change in its conformation [127,128]. Therefore, the destabilization of the monomeric α-syn conformation promotes aggregation and generates toxic species [129]. Moreover, post-translational modifications such as phosphorylation, O-GlcNAcylation, nitration, acetylation, ubiquitination, SUMOylation, and glycation in α-syn play a key role in its structure, function, and aggregation. Proteins with these modifications have been shown at the periphery of the Lewy body, and their levels have been found to be increased in the brains of PD patients. Interestingly, these modifications can alter glutamatergic signaling and exacerbate motor coordination and cognitive and olfactory dysfunction [130].

Another aspect of α-syn pathogenicity is related to the ability to spread, in its misfolded form, to different cell types and in different brain regions through anatomically connected networks. These effects, although not yet completely characterized in terms of factors and mechanisms involved, are also responsible for the appearance of disease-related clinical signs. Interestingly, it has been demonstrated that α-syn accumulation is correlated with more severe cognitive dysfunction [111,131]. Indeed, experiments performed in mice bearing mutations in α-syn display a reduction in cognitive performance time before the appearance of motor symptoms, thus confirming the relevance of identifying cognitive impairment as a prodromal hallmark of PD [132].

### 3.2. Mitochondrial Dysfunction

Mitochondria have multiple roles in cellular homeostasis. The main function of these organelles is to produce energy in the form of ATP, which is essential for cell proliferation and differentiation. However, mitochondria play a key role in intracellular calcium signaling because they finely tune the calcium cycle from the mitochondria to the cytosol and from the cytosol to the mitochondria [133,134]. This function is extremely important not only for regulating cytosolic calcium homeostasis but also for preserving the electrochemical gradient at the level of the inner mitochondrial membrane, thus ensuring mitochondrial energy production. Indeed, maintaining the Ca^2+^ concentration within a physiological range is an important requirement for ensuring mitochondrial function, since mitochondrial dehydrogenases, which are mainly involved in the regulation of oxidative phosphorylation and ATP synthesis [135], are dependent on calcium for their activities. Neurons in the brain have a high metabolic demand and, therefore, are tremendously dependent on mitochondrial efficiency. Indeed, in neurons, mitochondrial distribution is mainly dictated by the ATP consumption that occurs at the synaptic level or in correspondence with Ranvier’s nodes [136,137].

Over the last decade, alterations in mitochondrial function have been considered central to the pathogenesis of sporadic and familial forms of PD and have been included among the most common cell autonomic mechanisms leading to neuronal degeneration [138,139,140]. This hypothesis is also supported by the functional properties of dopaminergic neurons, which playing a pace-making role and are more prone to intracellular calcium transients. Consequently, mitochondria are exposed to calcium overload to improve mitochondrial dehydrogenase activity and free radical production. These events accelerate early dopaminergic neuronal aging [117].

Observations from experimental models of PD and human samples from patients with PD provide strong evidence that the disturbance of mitochondrial dynamics, bioenergetic defects, inhibition of the electron transport chain (ETC), complex I activity, and increased reactive oxygen species (ROS) production might be included among the pathogenetic factors underlying disease progression [141,142].

Interestingly, the impairment of mitochondrial function occurs with a complex spectrum of mechanisms, which, depending on the brain region involved, is associated with different patterns of cognitive damage [11,19] (Table 1). In this regard, reduced ATP production that is detected in the hippocampus and prefrontal cortex is associated with memory deficits [11]. On the other hand, impairments in mitochondrial dynamics and morphology, which occur in the parietal and occipital cerebral cortex and basal ganglia, may cause visual spatial impairments and slowed cognitive processing [13,14]. In addition, mutations in mt-DNA and the accumulation of damaged mt-DNA in neurons under oxidative stress conditions in different brain regions, like the prefrontal cortex, medial frontal gyrus, parietal lobe, and limbic system, cause attention deficits, depression, anxiety, and motor and cognitive impairments [16,143,144]. Interestingly, mutated mt-DNA may be subsequently released outside the neurons, thus contributing to the spread of PDD in an “infectious-like” manner [144].

Other types of mitochondrial dysfunctions, which occur within the cerebral cortex and basal ganglia, are (a) the impairment of complex I function, (b) the reduction in the electron transport activity, (c) the decrease in membrane potential, and (d) oxidative stress. All these events are associated with executive dysfunctions, impairments in problem solving abilities, and multitasking difficulties [12,143,145]. Moreover, α-syn binding to the inner mitochondrial membrane interferes with complex I activity, leading to mitochondrial dysfunction and increased mitophagy [146,147,148]. Therefore, α-syn overexpression/oligomerization at the level of mitochondrial complex I exerts a causative role in the onset of PD [149,150]. In addition, it has been found that α-syn accumulation within mitochondria causes an increase in intra-mitochondrial Ca^2+^ levels [115,116], which, in turn, leads to the increase in NO levels, oxidative damage, and cytochrome C release [146], all of which are intracellular events that contribute to the promotion of apoptotic neuronal death [151]. Therefore, a vicious circle arises in which α-syn aggregation and mitochondrial dysfunction exacerbate each other. This explains why both these cellular events are detectable and relevant in the degenerating dopaminergic and non-dopaminergic neurons in PD patients [108,152]. It is worth pointing out that α-syn accumulation and mitochondrial functional impairment mainly occurring at presynaptic terminals represent neuropathological features that are detectable not only at the nigrostriatal level but also in other brain regions involved in cerebral cognition, such as the prefrontal cortex, locus coeruleus, and the central nucleus of the amygdala [78,153].

Finally, α-syn has been found to have a non-canonical mitochondrial-targeting sequence and has been shown to influence mitochondrial structure and function [154].

All these findings further support the tight relationship between mitochondrial dysfunction and α-syn as a pathogenetic mechanism in the cellular dysfunction that characterizes PD and may help us understand the cellular and molecular mechanisms leading to the motor and non-motor symptoms related to the disease. Moreover, these considerations further underline the importance of studying the relationship between the molecular mechanisms involved in mitochondrial functional impairment and deficit in brain cognition as a prodromal marker of early neuronal damage. This would be useful to delineate a common intervention strategy to halt PD progression. This aim became extremely important to have consistent models to validate these parameters as diagnostic and prognostic markers of disease evolution, as well as to identify new druggable targets. This would help to overcome the gap existing, sometimes, in the translatability from preclinical to clinical studies. In this regard, to identify reliable biomarkers for the early detection and monitoring of cognitive impairment, it might be extremely useful.

On the other hand, it is important to emphasize that mitochondrial dysfunction represents a detrimental condition not only for neurons but also for glial cells, considering the role of astrocytes in the regulation of synaptic plasticity, as well as their trophic support on neuronal populations in the cerebral parenchyma. Specifically, microglial cells play a crucial role in the dynamic homeostasis of the brain microenvironment, and neuroinflammation caused by microglial activation has received increasing attention as an additional pathogenetic determinant in PD. Notably, activated microglia display two different phenotypes, one with pro-inflammatory features and the other with anti-inflammatory features. Both phenotypes have different metabolic patterns that help the cells to adapt to their functional characteristics [155]. Interestingly, mitochondria may play a role in this phenomenon because of their ability to continuously change their morphology according to cellular demands. Indeed, mitochondrial fission removes dysfunctional mitochondria and is responsible for the transition of microglia from OXPHOS to glycolysis; whereas mitochondrial fusion increases the number of mitochondrial cristae and promotes OXPHOS [156]. These effects are responsible for the polarization of microglial cells and, consequently, for the switch from a beneficial to a detrimental phenotype. In addition, mitophagy has been reported to contribute to the elimination of dysfunctional mitochondria, prompting the transformation of cells to a pro-inflammatory and glycolysis-dependent beneficial phenotype [157], whereas enhanced mitochondrial biogenesis and mitochondrial cellular content can induce cell polarization towards the detrimental phenotype [158,159]. These findings assume an important relevance considering that neuroinflammation may contribute to the development of different patterns of cognitive dysfunction depending on the brain regions involved, as recently reported [160,161,162].

## 4. Treatment Strategies in PD: Unsatisfied Medical Needs

Currently, the treatment of Parkinson’s disease is focused on reducing the severity of motor and non-motor symptoms, but there are no drugs that can cure or slow down the progression of PD (i.e., Disease-Modifying Drugs, DMDs) and the accompanying cognitive impairment, which still represents an unmet medical need [70].

Treatment with levodopa or dopamine receptor agonists helps to alleviate motor symptoms, whereas treatments for cognitive and neuropsychiatric symptoms are underdeveloped and provide poor results. The treatment of non-motor symptoms requires the use of non-dopaminergic drugs, such as cholinesterase inhibitors for cognitive decline and selective serotonin reuptake inhibitors for mental deterioration, but other pharmacological and non-pharmacological strategies need to be tested. Moreover, trihexyphenidyl, an anticholinergic drug, and amantadine, an antiglutamatergic drug, are used in patients with PD [163], and different formulations of ropinirole, pramipexole, and rotigotine are used during the initial phase or along with levodopa or carbidopa during the later phase of therapy.

Most of these drugs are administered orally; however, few can be administered through other routes of administration, such as transdermal, as occurs for rotigotine, or through intraduodenal infusion, as reported for levodopa [164], in order to allow their continuous delivery [163]. On the other hand, adenosine receptor antagonists, coenzyme Q10, creatine, isradipine, protein aggregation inhibitors, stem cells, and gene therapy represent alternative strategies to treat PD in preclinical and clinical investigations [163]. Unfortunately, none of the current therapies approved for PD treatment can significantly improve cognitive symptoms without causing any adverse effects. Levodopa, for example, is reported to benefit spatial working memory but its chronic use causes abnormal movements, visual hallucinations, psychosis, and fluctuations in motor performance [165,166]. Similarly, rivastigmine, the only drug approved by the FDA for the treatment of cognitive decline associated with PD, causes nausea, vomiting, tremor, diarrhea, anorexia, and dizziness [167,168]. Conversely, donepezil, a well-known cholinesterase inhibitor, and memantine, an NMDA receptor antagonist, both of which are used in AD patients, have not yet been recommended for the treatment of dementia associated with PD [169]. In this scenario, the results of preclinical studies showing that mitochondria might play a key role in this pathological condition accompanying PD in approximately 60–80% of patients represents a new promising avenue to follow in order to identify selective therapeutic strategies to prevent cognitive decline in PD. This hypothesis is supported by the recent finding that mitochondrial dysfunction is emerging as a prominent element triggering cognitive deterioration in most PD patients, since mitochondrial dysfunction may influence cognitive progression over the time [170,171].

Therefore, mitochondria represent a highly promising target for the development of both PD biomarkers and neuroprotective compounds that are able to modulate mitochondrial function. In this regard, various strategies have been developed to improve mitochondrial function in PD [172]. Among them, antioxidant therapies, due to their ability to regulate different metabolic pathways, ameliorate mitochondrial function in a timely and effective manner and represent a promising strategy for treating PD. On the other hand, gene therapy pointing to key strategic targets in the mitochondrial pathway might be able to correct gene mutations that result in focal disease and prevent dopaminergic neuronal degeneration. More importantly, novel gene editing technologies like CRISPR-CAS9 are being implemented in this regard to alter the DNA of germ cells to protect offspring from familial PD [173]. Moreover, therapeutic approaches targeting mitochondrial biogenesis are emerging as potential novel neuroprotective strategies in PD. Interestingly, recent research demonstrated that dopamine D1 receptor agonists can ameliorate mitochondrial biogenesis in a rat model of PD [174]. It is worth mentioning that mitochondrial transplantation has been revealed as a promising strategy in PD also. Finally, pharmacological strategies aimed at enhancing mitophagy in dopaminergic neurons represent a meaningful therapeutic approach in PD [175]. Indeed, experimental models from iPSCs collected from PD patients with parkin or PINK1 mutations allowed researchers to identify at least four candidate compounds that are able to improve the clearance of damaged mitochondria [176]. Recently, the demonstration that damaged mtDNA plays a key role in neurotoxicity and in the spread of an infectious PDD-like pathology paves the way for innovative treatment strategies and monitoring approaches for PD [144]. Worth mentioning are also the results of preclinical studies demonstrating the beneficial effects of insulin-like growth factor 1 (IGF-1) on synaptic plasticity and cognitive functions in animal models reproducing the sporadic form of PD [177]. Moreover, among the novel strategies to modulate cognitive dysfunctions in PD, subthalamic nucleus stimulation (STN) has attracted considerable attention due to its ability to affect both cognitive and affective components of behavior in human beings, thus improving the social interaction deficits observed in PD [178].

## 5. Conclusions

In conclusion, cognitive impairment accompanying PD is a complex and multifaceted issue that significantly impacts patients and their families in a high percentage of PD cases. Although our understanding of the causes and mechanisms underlying cognitive impairment in PD has greatly advanced, much remains to be addressed. An important finding is related to its appearance before the manifestation of motor dysfunction. Moreover, the discovery that mitochondria play a key role in the pathogenesis of the cerebral damage responsible for cognitive impairment in PD that is associated with synucleinopathy has become extremely relevant in terms of diagnosis, prognosis, and therapy [179,180]. Identifying biomarkers to better predict cognitive decline, recognizing patients at a high risk of early and rapid cognitive deterioration, and developing disease-modifying therapies based on the improvement of mitochondrial function and activity of synaptic proteins involved in neurotransmitter release represent important achievements in the diagnosis and treatment of cognitive impairment associated with PD, as well as in predicting the fate of the patients affected by this detrimental pathology. In this regard, alongside clinical studies, it is essential to develop animal models that are highly predictive of PD progression. 

## Figures and Tables

**Figure 1 ijms-25-11490-f001:**
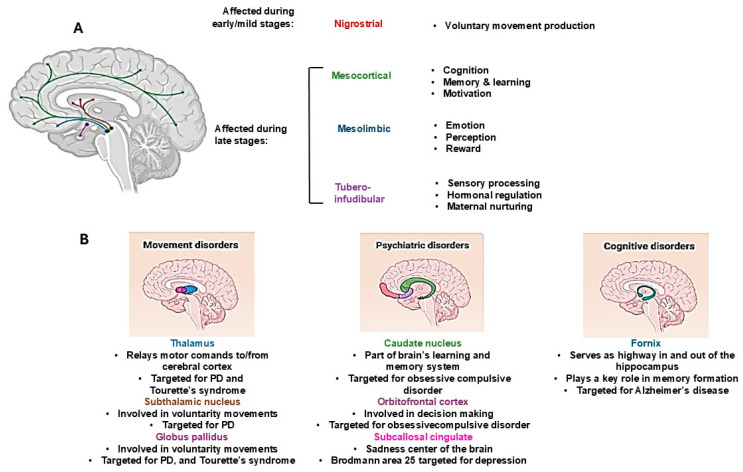
Dopaminergic pathways involved in motor and non-motor symptoms in Parkinson’s Disease. (**A**) Dopaminergic cerebral connections and their involvement in the regulation of brain function in physiological and pathological conditions; (**B**) dopaminergic brain regions affected in motor and non-motor disorders associated with PD.

**Figure 2 ijms-25-11490-f002:**
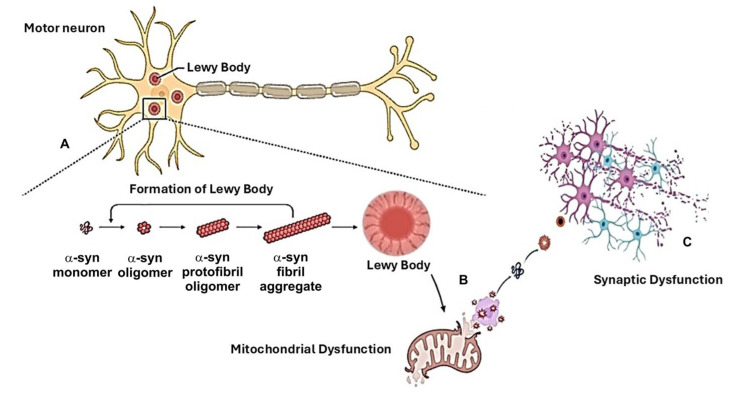
Relationship between α-syn aggregation and mitochondrial dysfunction as pathogenetic mechanism of neuroplasticity rearrangement. (**A**) The increased deposition of misfolded α-syn within the Lewy bodies (**B**) induces mitochondrial dysfunction in the soma and neurites of dopaminergic neurons, with (**C**) consequent alterations in synaptic communication and plasticity.

**Table 1 ijms-25-11490-t001:** Summary of the alterations in mitochondrial molecular mechanisms, brain regions, manifestations of cognitive impairment in PD, and the respective models used.

Mitochondrial Dysfunction	Brain Areas	Cognitive Impairment	Models	Ref.
Reduced ATP production	Hippocampus and prefrontal cortex	Memory deficits	Preclinical cellular models	[11]
Increased oxidative stress	Prefrontal cortex and basal ganglia	Executive dysfunction	Clinical models (observations in patients)	[12]
Impairment in mitochondrial dynamics (fission/fusion)	Parietal lobe and occipital lobe	Visuospatial impairments	Preclinical cellular models	[13]
Altered mitochondrial morphology	Cerebral cortex and basal ganglia	Slowed cognitive processing	Human post-mortem tissues	[14]
Impaired electron transport chain	Temporal lobe, frontal lobe	Language difficulties	Clinical models (patients)	[15]
Mutations in mitochondrial DNA	Prefrontal cortex and parietal lobe	Attention deficits	Preclinical animal models	[16]
Reduction in mitochondrial membrane potential	Frontal lobe	Impaired problem-solving abilities	Preclinical cellular and animal models	[17]
Accumulation of mitochondrial DNA damage	Limbic system and prefrontal cortex	Mood disorders (depression and anxiety)		[18]
Complex I deficiency of the electron transport chain	Prefrontal cortex and basal ganglia	Difficulties in multitasking		[19]

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
