# Peer review of "Mitochondrial Dysfunction in Parkinson’s Disease: A Contribution to Cognitive Impairment?"

_ijms, 2024, doi:10.3390/ijms252111490_

Round 1
Reviewer 1 Report (New Reviewer)
Comments and Suggestions for Authors
The manuscript " Cognitive impairment in Parkinson’s disease: Is this a question regarding mitochondria?" discusses about the intricate relationship between cognitive decline and Parkinson’s disease (PD) focusing on the underlying factors that contribute to the disorder and the brain regions involved. In addition, the role of mitochondria in synaptic plasticity has also been discussed with reference to established understandings. As of yet, several studies have elaborately explored on the connection between mitochondrial dysfunction and PD, especially the molecular triggers of the disease process and the intra- and extracellular roles of mitochondria in PD. The submitted manuscript further explores the involvement of mitochondrial dysfunction in PD.
After going through the manuscript, I have following comments for the authors.
1. Although the manuscript discusses on the relationship of mitochondrial dysfunctions with cognitive impairments in PD, i would suggest the authors to briefly discuss on the classical neurobiological basis for PD - degeneration of nigrostriatal dopamine neurons and the pathological deposition of the protein α-synuclein in intraneuronal Lewy inclusions within vulnerable populations of neurons in the brain.
2. Please briefly discuss on the genetic aspect of cognitive impairment in PD highlighting the contributions of PARK PD-associated genes.
Comments on the Quality of English LanguageMinor grammatical corrections and syntax adjustments recommended.
Author Response
Comment #1: Although the manuscript discusses on the relationship of mitochondrial dysfunctions with cognitive impairments in PD, I would suggest the authors to briefly discuss on the classical neurobiological basis for PD -degeneration of nigrostriatal dopamine neurons and the pathological deposition of the protein α-synuclein in intraneuronal Lewy inclusions within vulnerable populations of neurons in the brain.
Answer: We thank the referee for this request, therefore we added a brief discussion about the classical neurobiological basis of PD in the revised version of the manuscript. Please, see page: 2, lines: 30-38 “From a neuropathological point of view, PD is characterized by the demise of dopaminergic (DA) neurons in the substantia nigra pars compacta (SNc) and the reduction of (DA) tone at the level of the striatum [2]. ………However, PD is characterized by more widespread pathology in other brain regions and involves non-dopaminergic neurons as well.”.
Comment #2: Please briefly discuss on the genetic aspect of cognitive impairment in PD highlighting the contributions of PARK PD-associated genes.
Answer: According with referee’s criticisms a brief discussion on the genetic aspect of cognitive impairment in PD highlighting the contributions of PARK PD-associated genes has been provided in the revised version of the manuscript. Please, see section 2, sub-section 2.1, pages: 5 and 6, lines: 214-227, “Finally, it cannot be overlooked the growing bulk of evidences related to the contribution of genetics to cognitive dysfunction in PD.............These variabilities in the genetic forms of PD further contribute to explain the different cognitive involvements in each form of genetic PD”; sub-section 2.3 page 8, lines: 325-328, “Moreover, other genes ……developing of cognitive decline”; and section 3 page 9, lines: 363-365 “is also supported by the pathophysiological features displayed by genetic forms of PD, such as those associated LRRK2, and PRKN and to a-synuclein (SNCA) genes mutations, in which cognitive impairment is also described”
Reviewer 2 Report (New Reviewer)
Comments and Suggestions for Authors
Although the topic of the manuscript is very important in it fits the scope of the journal, I believe that major changes are necessary to enhance the manuscript's clarity and scientific rigor. These changes are critical to ensure that the manuscript provides a cohesive, accurate, and comprehensive examination of the current understanding of PD.
Abstract: What is the main goal of the manuscript? What are the main conclusions?
Punctuation should be added in sentences. Several sentences are long and lack proper punctuation. The introduction should end at line 54, where the authors present the purpose of the manuscript. However, before stating the purpose, the authors should add 1 or 2 sentences about the importance of mitochondria in Parkinson's disease (PD).
In Table 1, the model used for each alteration should be added, or, if the observations are from humans, it should be noted if they are from post-mortem tissue.
What kinds of cognitive deficits are observed in PD patients? A section about this should be added to the manuscript.
In the section about mitochondrial dysfunction, the authors should focus specifically on alterations detected in PD. They describe the role of mitochondria in the M1 and M2 microglia phenotypes but do not discuss the importance of microglia phenotypes for PD. In fact, the references used are not related to PD but to other diseases. This should be changed and rewritten.
Why does alpha-synuclein become pathogenic? What factors cause alpha-synuclein to oligomerize? This should be added and discussed. Another point is that the section on alpha-synuclein should come first, followed by mitochondrial dysfunction.
The sequence of the sections should be revised. In my view, the first topic should be the characterization of PD in terms of cognitive decline, followed by the animal models of PD, and then the pathophysiological mechanisms of the disease. Another important point is that the title of the manuscript does not accurately represent its content.
Author Response
Comment #1: Abstract: What is the main goal of the manuscript? What are the main conclusions?.
Answer: As requested by the Referee the abstract has been modified indicating the main goal of the manuscript and the main conclusions. Please, see page: 1, lines: 15-22: “The aim of the present review is …………..with attention to mitochondria and their contribution to synaptic dysfunction and neuronal deterioration in the brain regions responsible for non-motor manifestations of the disease”.
Comment #2: Punctuation should be added in sentences. Several sentences are long and lack proper punctuation. The introduction should end at line 54, where the authors present the purpose of the manuscript. However, before stating the purpose, the authors should add 1 or 2 sentences about the importance of mitochondria in Parkinson's disease (PD).
Answer: According with Referee’s criticisms the introduction has been modified and the punctuation added in the sentences. Please, see section 1, pages: 1-3. Moreover, as requested by the Referee, two sentences about the importance of mitochondria in PD have been added before stating the purpose. Please see page 2, lines: 54-66 “In this regard mitochondrial dysfunction, a mechanistic condition related to prodromal symptoms leading to early deficits in cognition in PD patients, .........., in many brain regions involved in the control of cognition, attention, mood, memory and executive functions as reported in table 1”.
Comment #3: In Table 1, the model used for each alteration should be added, or, if the observations are from humans, it should be noted if they are from post-mortem tissue.
Answer: Table 1 has been modified according to referee’s suggestion. Therefore, in the revised version of the manuscript a column has been inserted in table 1 to indicate the models used to describe each alteration reported. Please see page 3 lines 96-98
Comment #4: What kinds of cognitive deficits are observed in PD patients? A section about this should be added to the manuscript.
Answer: We thank the referee for his/her suggestion to indicate the type of cognitive deficits observed in PD patients. Therefore, in the revised version of the manuscript this criticism has been addressed. Please see section 2.1 Clinical symptoms of cognitive dysfunctions, pages 2, lines: 79-91 “The clinical manifestations of cognitive impairment associated to PD includes a wide spectrum of symptoms ranging from........ or making moral decisions”; page 3, lines: 99-108 “Boller and co-workers found also deficits in both visuoperceptive and visuomotor tasks …………Other symptoms that coexist in PD patients with cognitive impairment include de-pression, hallucinations, and sleep disturbances”
Comment #5: In the section about mitochondrial dysfunction, the authors should focus specifically on alterations detected in PD. They describe the role of mitochondria in the M1 and M2 microglia phenotypes but do not discuss the importance of microglia phenotypes for PD. In fact, the references used are not related to PD but to other diseases. This should be changed and rewritten.
Answer: We apologize for the misunderstanding related to the description of the role of mitochondria in determining microglial phenotype. Our aim was to underline the relevance of mitochondrial dysfunctions in determining cell fate also in non neuronal cells like glial cells that exert both beneficial and detrimental effects in brain parenchyma. We agree with the referee about the importance of microglia phenotypes for PD, however, the topic is so complex and noteworthy that should deserve a specific discussion that is out of scope of this review.
Comment #6: Why does alpha-synuclein become pathogenic? What factors cause alpha-synuclein to oligomerize? This should be added and discussed. Another point is that the section on alpha-synuclein should come first, followed by mitochondrial dysfunction.
Answer: We thank the referee for this important suggestion to discuss the mechanisms involved in the pathogenicity of alpha synuclein and its oligomerization. Therefore, in the revised version of the manuscript a brief description of these mechanisms has been provided. Please, see section 3, subsection 3.1, page: 10, lines: 410-420 “In this regard, it is important to underline the role of cellular environments ….. these modifications can alter glutamatergic signaling and exacerbate motor coor-dination and cognitive and olfactory dysfunction”. Moreover, as suggested by the Referee the section on alpha-synuclein has been moved before the section on mitochondrial dysfunction. Please see section 3.1 Role of a-syn, pages 10-11, lines 377-429 …
Comment #7: The sequence of the sections should be revised. In my view, the first topic should be the characterization of PD in terms of cognitive decline, followed by the animal models of PD, and then the pathophysiological mechanisms of the disease. Another important point is that the title of the manuscript does not accurately represent its content.
Answer: The sequence of the sections has been revised according the referee’s suggestion. Therefore, in the revised version of the manuscript the first topic corresponds to the characterization of PD in terms of cognitive decline, see section 2 (2.1 and 2.2) pages 2-7, lines: 77-280, “Cognitive impairment in PD”, followed by the animal models of PD, see section 2.3, pages 7-9, lines 281-343, “Cognitive impairment in animal models of PD” followed by the pathophysiological mechanisms of the disease, see section 3, pages 9-12, lines: 344-527, “α-synuclein and mitochondrial involvement in the pathogenetic mechanisms related to cognitive decline in PD”.
Finally, the title of the manuscript has been changed as follow: “Mitochondrial dysfunction in Parkinson’s disease: a contribution to cognitive impairment?”
Reviewer 3 Report (New Reviewer)
Comments and Suggestions for Authors
In their review article entitled “Cognitive impairment in Parkinson’s disease: Is this a question regarding mitochondria?” the Authors set out to evaluate cognitive disorders in Parkinson's disease, focusing mainly on mitochondrial dysfunction as the underlying molecular mechanism. The idea behind the manuscript is interesting; however, some points should be addressed.
The message the authors want to send falls more into the category of a perspective article than a review.
Mitochondrial dysfunction could be one of the molecular causes at the basis of the synaptic failure and disconnection that underlies the non-motor symptoms of PD, however, the present title overemphasis the concept, I would propose to change it as follows: Mitochondrial dysfunction in Parkinson’s disease: a contribution to cognitive impairment?
I would suggest eliminating the bulleted list present in the body of the abstract and briefly summarizing the points, concluding with the purpose of the review.
I would suggest moving paragraph 3 with the description of cognitive deficits to the first paragraph after the introduction. I would also suggest summarizing the paragraph “Clinical studies” (lines 257 to 305) which appears as a simple introduction to section 3.1.2 “Clinical symptoms of cognitive dysfunction.”
Paragraph 3.2, the title should be changed to read as follows: “Cognitive impairment in animal models of PD.” In addition, among the various animal models mentioned in the text, I would suggest, to add novelty to the manuscript, that the most modern models of synucleinopathy induced by intracerebral injection of alpha-synuclein aggregates be emphasized, as they represent early models of the pathology and an excellent means of studying non-motor prodromal symptoms such as cognitive deficits.
Paragraph 2 is titled “Pathogenetic mechanisms related to cognitive decline” however, there is no description of the different hypothesized mechanisms but a single focus on mitochondrial dysfunction. Therefore, I would suggest changing the title of the paragraph by sticking to the content or changing the content by adapting it to the title. Adapting the subchapters accordingly as well.
In particular, for subchapter 2.1, I recommend the evaluation of the present works:
PMID: 37779111
PMID: 39102941
For subchapter 2.2, when the authors talk about the detrimental effects of alpha-synuclein at the synaptic level, I recommend the evaluation of the following works, some of which are worth mentioning.
PMID: 26392130
PMID: 30927362
PMID: 34297092
PMID: 36414406
Regarding the last paragraph “Treatment Strategies in PD: Unsatisfied medical needs,” I would recommend further literature search and citation of the following recent works on the topic.
PMID: 38696281
PMID: 38750702
Comments on the Quality of English Language
There are some typos in the text, I recommend a language check.
Author Response
Comment #1: Mitochondrial dysfunction could be one of the molecular causes at the basis of the synaptic failure and disconnection that underlies the non-motor symptoms of PD, however, the present title overemphasis the concept, I would propose to change it as follows: Mitochondrial dysfunction in Parkinson’s disease: a contribution to cognitive impairment?
Answer: According the Referee’s suggestion the title of the manuscript has been changed. Therefore, in the revised version the title is: “Mitochondrial dysfunction in Parkinson’s disease: a contribution to cognitive impairment?”
Comment #2: I would suggest eliminating the bulleted list present in the body of the abstract and briefly summarizing the points, concluding with the purpose of the review.
Answer: According to the Referee’s suggestion the abstract has been modified. Therefore, in the revised version of the manuscript the bullet list present in the body of the abstract has been eliminated, the points summarized and the purpose of the study indicated in the conclusions. Please, see Please, see page: 1, lines: 15-22: “The aim of the present review is …………..with attention to mitochondria and their contribution to synaptic dysfunction and neuronal deterioration in the brain regions responsible for non-motor manifestations of the disease”.
Comment #3: I would suggest moving paragraph 3 with the description of cognitive deficits to the first paragraph after the introduction. I would also suggest summarizing the paragraph “Clinical studies” (lines 257 to 305) which appears as a simple introduction to section 3.1.2 “Clinical symptoms of cognitive dysfunction.”
Answer: According to the referee’s suggestion the paragraph on cognitive deficits has been moved after the introduction. Moreover, the paragraph has been rearranged and the titles of the sub-paragraphs modified as suggested. Please, see answer to comment # 7 of Referee 2
Comment #4: Paragraph 3.2, the title should be changed to read as follows: “Cognitive impairment in animal models of PD.” In addition, among the various animal models mentioned in the text, I would suggest, to add novelty to the manuscript, that the most modern models of synucleinopathy induced by intracerebral injection of alpha-synuclein aggregates be emphasized, as they represent early models of the pathology and an excellent means of studying non-motor prodromal symptoms such as cognitive deficits.
Answer: According to the Referee’s suggestion the title of paragraph 3.2, now 2.3 in the revised version of the manuscript, has been changed as follow: “Cognitive impairment in animal models of PD”. Please, see page: 7 line: 281. Moreover, to accomplish referee’s request, among the models reported in this paragraph, the most modern models of synuceinopathy induced by injection of alpha-synuclein aggregates have been emphasized. Please, see page 8, lines: 302-309, “In addition, new generation of animal models of PD have been developed by inoculating α-syn pre-formed fibrils (PFF) in the substantia nigra or the striatum of rodents [90]. These models, mimicking the ectopic expression or overexpression of the protein α-syn, allow to replicate the pathological hallmarks of PD (Fig. 2). Indeed, the use of these models, importantly contributed to study the role of toxic forms of α-syn in neurodegeneration, and their influence in the processes occurring before cell death, such as changes in synaptic transmission, basal ganglia plasticity and mechanisms underlying motor learning”
Comment #5: Paragraph 2 is titled “Pathogenetic mechanisms related to cognitive decline” however, there is no description of the different hypothesized mechanisms but a single focus on mitochondrial dysfunction. Therefore, I would suggest changing the title of the paragraph by sticking to the content or changing the content by adapting it to the title. Adapting the subchapters accordingly as well.
Answer: According to the Referee’s suggestion the title of paragraph 2, now 3 in the revised version of the manuscript, has been changed as follow: “α-synuclein and mitochondrial involvement in the pathogenetic mechanisms related to cognitive decline in PD”. Please, see page 9, lines 344-345.
Comment #6: In particular, for subchapter 2.1, I recommend the evaluation of the present works: PMID: 37779111, PMID: 39102941. For subchapter 2.2, when the authors talk about the detrimental effects of alpha-synuclein at the synaptic level, I recommend the evaluation of the following works, some of which are worth mentioning: PMID: 26392130, PMID: 30927362, PMID: 34297092, PMID: 36414406.
Answer: As suggested by the Referee the paper recommended have been included and discussed in the text, paragraph 3.1 and 3.2 of the revised version of the manuscript. Please see paragraph 3.1, page 10, lines 385-389, “ This last finding is dependent on the interference of α-syn with glutamatergic neurotransmission not only at the level of dopaminergic striatal terminals, where it reduces dopamine release, but also at level of postsynaptic density of medium spiny neurons by interfering with different subunits of NMDA receptors respectively [119–124]”, and 393-398 “ Hallam and collaborators demonstrated that the impairment in NMDAR activity, caused by aberrant insoluble and phosphorylated a-syn deposits at synaptic terminals, temporally correlates with increased nitric oxide synthesis and S-nitrosylation of the dendritic scaffold protein, microtubule-associated protein 1A [125]. These data suggest that loss of synaptic function in Lewy body dementia may result from synucleinopathy-evoked nitrosative stress and subsequent NMDAR dysfunction [125].” For paragraph 3.2 please see page 11, lines 465-475, “In addition, mutations in mt-DNA and accumulation of damaged mt-DNA occurring in neurons, under oxidative stress conditions, in different brain regions like prefrontal cortex, medial frontal gyrus, parietal lobe and limbic system, cause attention deficits, depression, anxiety, motor and cognitive impairments [16,143,144]. Interestingly, mutated mt-DNA may be subsequently released outside the neurons thus contributing to the spreading of PDD in an “infectious-like” manner [144]. Other types of mitochondrial dysfunctions, which occur within cerebral cortex and basal ganglia, are (a) the impairment of complex I function, (b) the reduction of the electron transport activity, (c) the decrease in membrane potential and (d) the oxidative stress. All these events are associated to executive dysfunctions, impairments in problem solving abilities and multitasking difficulties [12,158,143,145].”
Comment #7: Regarding the last paragraph “Treatment Strategies in PD: Unsatisfied medical needs,” I would recommend further literature search and citation of the following recent works on the topic: PMID: 38696281,PMID: 38750702.
Answer: According to Referee’s suggestion the manuscripts recommended have been included in the paragraph 4 of the revised version of the manuscript. Please see pages 13-14, lines 583-592, “Recently, the demonstration that damaged mtDNA plays a key role in neurotoxicity and in the spread of an infectious PDD-like pathology paves the way for innovative treatment strategies and monitoring approaches for PD[144]. Worth of mention are also the results of preclinical studies demonstrating the beneficial effects of insulin-like growth factor 1 (IGF-1) on synaptic plasticity and cognitive functions in animal models reproducing sporadic form of PD [177]. Moreover, among the novel strategies to modulate cognitive dysfunctions in PD, subtalamic nucleus stimulation (STN) has attracted considerable attention due to its ability to affect both cognitive and affective components of behaviour in human beings thus improving the social interaction deficits observed in PD [178]”
Round 2
Reviewer 2 Report (New Reviewer)
Comments and Suggestions for Authors
The authors significantly improved the manuscript in the revised form. I recommend the publication.
This manuscript is a resubmission of an earlier submission. The following is a list of the peer review reports and author responses from that submission.
Round 1
Reviewer 1 Report
Comments and Suggestions for Authors
In this review, Sirabella and colleagues discuss the relevance of cognitive impairment as a prevalent and debilitating non-motor symptom in Parkinson's Disease (PD). They highlight the range of cognitive issues from early to late stages of PD, emphasizing the need for a deeper understanding of the molecular mechanisms behind cognitive decline in PD, with a particular focus on mitochondrial dysfunction and its role in synaptic and neuronal deterioration. In this sense, the review aims to explore these pathogenetic mechanisms to aid in the differential diagnosis, prognosis, and development of therapeutic strategies for cognitive impairment in PD.
General comments:
The topic is relevant and the review is well-written and easy to follow. However, in section 3 [Pathogenetic mechanism related to cognitive decline in PD: role of Alpha-synuclein (α-syn) and mitochondria], the authors could better describe the connection between mitochondria, cognitive decline, and PD. Currently, the text generally describes the well-established relationship between PD and mitochondria dysfunction. Additionally, the review could benefit from a more explicit discussion of how the understanding of the molecular mechanisms related to cognitive impairment in PD might translate into clinical practice, particularly in terms of diagnostic and prognostic guidelines.
Including a summary table of the mitochondrial molecular mechanisms related to cognitive impairment in PD and their original references would also help to enrich the content, making it more informative for researchers and clinicians.
Minor comments:
- Figure 1: legend should be more descriptive since the image involves movement, psychiatry and cognitive disorders.
- Lines 173-176: it would be beneficial if the authors could provide a more detailed discussion on the causes of transient cognitive disturbances in PD patients, such as infections, metabolic and electrolyte imbalances (e.g., hypoglycemia, hypercalcemia, high calcium levels, hyponatremia), thyroid disorders, dehydration, nutritional deficiencies (e.g., vitamin B12 and thiamine deficiency), sleep disorders, and chronic fatigue.
- Line 201: “such as those caused by LRRK2 and A53T mutations”. Rephrase the sentence using appropriate nomenclature for genes and variants.
- Figure 2: The legend should be more self-explanatory.
Comments on the Quality of English LanguageThe quality of English language is ok.
Author Response
Comment #1: The topic is relevant and the review is well-written and easy to follow. However, in section 3 [Pathogenetic mechanism related to cognitive decline in PD: role of Alpha-synuclein (α-syn) and mitochondria], the authors could better describe the connection between mitochondria, cognitive decline, and PD. Currently, the text generally describes the well-established relationship between PD and mitochondria dysfunction. Additionally, the review could benefit from a more explicit discussion of how the understanding of the molecular mechanisms related to cognitive impairment in PD might translate into clinical practice, particularly in terms of diagnostic and prognostic guidelines.
Answer: We thank the referee for these important observation that has been addressed in the revised version of the manuscript both in section 3 [Pathogenetic mechanism related to cognitive decline in PD: role of Alpha-synuclein (α-syn) and mitochondria], and in section 1 [Introduction]. Please, see pages: 3 and 4, lines 60-100 and page: 11, lines 416-422.
Comment#2: Including a summary table of the mitochondrial molecular mechanisms related to cognitive impairment in PD and their original references would also help to enrich the content, making it more informative for researchers and clinicians.
Answer: according to referee request a table summarizing the mitochondrial molecular mechanisms related to cognitive impairment has been included in the text. Please, see page… of the revised version of the manuscript. Please see page 3.
Comment #3: Figure 1: legend should be more descriptive since the image involves movement, psychiatry and cognitive disorders
Answer: We apologize for the lack of explanation in the legend of Fig1. In the revised version of the manuscript the legend of Fig 1 is more descriptive. Please see page: 5, lines: 147-149
Comment #4: Lines 173-176: it would be beneficial if the authors could provide a more detailed discussion on the causes of transient cognitive disturbances in PD patients, such as infections, metabolic and electrolyte imbalances (e.g., hypoglycemia, hypercalcemia, high calcium levels, hyponatremia), thyroid disorders, dehydration, nutritional deficiencies (e.g., vitamin B12 and thiamine deficiency), sleep disorders, and chronic fatigue.
Answer: according to referee request a more detailed discussion on the causes of transient cognitive disturbance in PD patients has been included in the revised version of the manuscript. Please see Section 2 of the new manuscript page: 7, lines: 229-269
Comment #5: Line 201: “such as those caused by LRRK2 and A53T mutations”. Rephrase the sentence using appropriate nomenclature for genes and variants.
Answer: We apologize for missing appropriate nomenclature in the sentence indicated by the referee. In the revised version of the manuscript the sentence has been rephrased properly. Please see Section 3 page: 8 lines: 294-295.
Comment #6: Figure 2: The legend should be more self-explanatory.
Answer: We apologize for the lack of explanation in the legend of Fig 2. In the revised version of the manuscript the legend of Fig 2 is more descriptive. Please see page: 8, lines: 300-303
Reviewer 2 Report
Comments and Suggestions for Authors
The manuscript addresses a critical and emerging topic in the field of Parkinson's Disease (PD) studies, focusing on cognitive impairment and its potential link to mitochondrial dysfunction. This is highly relevant given the growing recognition of non-motor symptoms in PD and their impact on patient quality of life. The manuscript's overall structure is logical, and the flow of information is clear. This study is commendable and deserves support. The following recommendations are provided for authors' reference:
1. The abstract provides a good manuscript overview but could be more concise. Consider summarizing key findings and implications in a few more precise sentences.
2. The review would benefit from a clearer distinction between the types of studies reviewed (e.g., clinical studies, animal models, in vitro studies) and a brief discussion of these studies' methodological quality and limitations.
3. The introduction sets the stage well but could be improved by briefly mentioning the potential therapeutic implications of understanding the link between cognitive impairment and mitochondrial dysfunction in PD. The following article may provide inspiration for the author, but the references should not be limited to those listed: Briston, Thomas, and Amy R. Hicks. "Mitochondrial dysfunction and neurodegenerative proteinopathies: mechanisms and prospects for therapeutic intervention." Biochemical Society Transactions 46.4 (2018): 829-842.; Yu & Wu. Mild cognitive impairment in patients with Parkinson's disease: An updated mini-review and future outlook. Front Aging Neurosci. 2022 Sep 6;14:943438.
4. For the "cognitive impairment in PD" section in the introduction, it is recommended that the author briefly introduce the topic of cognitive function assessment in PD and another emerging cognitive domain—social cognitive function. This will enhance the comprehensiveness and completeness of the study's description of cognitive function. The following references are provided for the author's consideration but are not limited to these:
-Assessment for PD cognitive function: Koevoets, Emmie W., Ben Schmand, and Gert J. Geurtsen. "Accuracy of two cognitive screening tools to detect mild cognitive impairment in Parkinson's disease." Movement Disorders Clinical Practice 5.3 (2018): 259-264.; Ya-Wen, et al. "A new instrument combines cognitive and social functioning items for detecting mild cognitive impairment and dementia in Parkinson’s disease." Frontiers in Aging Neuroscience 14 (2022): 913958.; Yu et al. (2020). Evaluating mild cognitive dysfunction in patients with Parkinson’s disease in clinical practice in Taiwan. Scientific reports, 10(1), 1014.;
-Social cognition domain: Czernecki, Virginie, et al. "Social cognitive impairment in early Parkinson's disease: A novel “mild impairment”?." Parkinsonism & related disorders 85 (2021): 117-121.; Kawamura, Mitsuru, and Shinichi Koyama. "Social cognitive impairment in Parkinson's disease." Journal of Neurology 254 (2007): IV49-IV53..; Yu et al. "Emotion-specific affective theory of mind impairment in Parkinson’s disease." Scientific Reports 8.1 (2018): 16043.
5. Pathogenetic Mechanism section is comprehensive but dense. Breaking it into smaller subsections with clear subheadings (e.g., "Role of α-synuclein", "Mitochondrial Dysfunction") could enhance readability.
6. The section on treatment strategies provides a good overview but could be enhanced by discussing emerging therapies and ongoing clinical trials in more detail.
Author Response
Comment #1: The abstract provides a good manuscript overview but could be more concise. Consider summarizing key findings and implications in a few more precise sentences.
Answer: We agree with referee’s criticism about the abstract conciseness. Therefore, in the revised version of the manuscript the abstract has been modified summarizing key findings and implications in a few precise sentences. Please, see page: 1, lines: 19-24
Comment #2: The review would benefit from a clearer distinction between the types of studies reviewed (e.g., clinical studies, animal models, in vitro studies) and a brief discussion of these studies' methodological quality and limitations.
Answer: According with referee’s criticisms a clearer distinction between the type of studies discussed in the manuscript has been provided underlining the methodological quality and limitations. Please, see section 2, page: 4, lines 124-126
Comment #3: The introduction sets the stage well but could be improved by briefly mentioning the potential therapeutic implications of understanding the link between cognitive impairment and mitochondrial dysfunction in PD.
Answer: We thank the referee for the precious suggestion to mention the therapeutic implications of understanding the link between cognitive impairment and mitochondrial dysfunction in PD. Therefore, in the revised version of the manuscript this aspect has been deeply addressed. Please, see section 1 pages: 3 and 4, lines 60-100 and page: 11, lines 416-422.
Comment #4: For the "cognitive impairment in PD" section in the introduction, it is recommended that the author briefly introduce the topic of cognitive function assessment in PD and another emerging cognitive domain—social cognitive function. This will enhance the comprehensiveness and completeness of the study's description of cognitive function.
Answer: we agree with referee’s suggestion that a brief discussion on the emerging cognitive domain related to social cognitive function might enhance the comprehensiveness and completeness of the study's description of cognitive function. Therefore, in the revised version of the manuscript this topic has been addressed. Please, see section 2 page:4, lines: 113-115 and 134-143.
Comment #5: Pathogenetic Mechanism section is comprehensive but dense. Breaking it into smaller subsections with clear subheadings (e.g., "Role of α-synuclein", "Mitochondrial Dysfunction") could enhance readability.
Answer: According to referee suggestion section 3 has been divided into two small subsection whit subheadings: "Role of α-synuclein", "Mitochondrial Dysfunction”. Please, see page: 8, lines: 308 and 350.
Comment #6: The section on treatment strategies provides a good overview but could be enhanced by discussing emerging therapies and ongoing clinical trials in more detail.
Answer: we thank the referee for the suggestion to discuss the emerging therapies and ongoing clinical trials in more detail. Therefore, in the section 4 of the revised version of the manuscript the discussion about new therapeutic strategies aimed to ameliorate the mitochondrial function, and their stage of clinical developing, have been discussed. Please, see page:13, lines: 517-533

Round 2
Reviewer 2 Report
Comments and Suggestions for Authors
The authors addressed each question briefly, but the content did not significantly improve. Additionally, the instructions in the reply letter did not correspond to the yellow markings in the manuscript, further adding to the confusion. For example, page 4, lines 124-126; lines 416-422.....
Author Response
Comment # 1: The authors addressed each question briefly, but the content did not significantly improve. Additionally, the instructions in the reply letter did not correspond to the yellow markings in the manuscript, further adding to the confusion. For example, page 4, lines 124-126; lines 416-422.....
Answer: We apologize for the incorrect correspondence of the yellow markings in the revision 1 of the manuscript. We corrected this mistake and, according to referee’s suggestion, we provided additional information about the type of studies reviewed i.e. clinical and preclinical studies. All these changes are now inserted in the revised version of the manuscript. Please see: page: 4, lines 120-133; lines: 139-143; page: 10, lines: 422-434
